# Flow Structures on a Planar Food and Drug Administration (FDA) Nozzle at Low and Intermediate Reynolds Number

**Adrián Corrochano** [1,*], **Donnatella Xavier** [2], **Philipp Schlatter** [2], **Ricardo Vinuesa** [2] and **Soledad Le Clainche** [1]

1   School of Aerospace Engineering, Universidad Politécnica de Madrid, 28040 Madrid, Spain; soledad.leclainche@upm.es
2   SimEX/FLOW, Engineering Mechanics, KTH Royal Institute of Technology, SE-100 44 Stockholm, Sweden; dgxavier@kth.se (D.X.); pschlatt@mech.kth.se (P.S.); rvinuesa@mech.kth.se (R.V.)
*   Correspondence: adrian.corrochanoc@alumnos.upm.es

**Abstract:** In this paper, we present a general description of the flow structures inside a two-dimensional Food and Drug Administration (FDA) nozzle. To this aim, we have performed numerical simulations using the numerical code Nek5000. The topology patters of the solution obtained, identify four different flow regimes when the flow is steady, where the symmetry of the flow breaks down. An additional case has been studied at higher Reynolds number, when the flow is unsteady, finding a vortex street distributed along the expansion pipe of the geometry. Linear stability analysis identifies the evolution of two steady and two unsteady modes. The results obtained have been connected with the changes in the topology of the flow. Finally, higher-order dynamic mode decomposition has been applied to identify the main flow structures in the unsteady flow inside the FDA nozzle. The highest-amplitude dynamic mode decomposition (DMD) modes identified by the method model the vortex street in the expansion of the geometry.

**Keywords:** FDA nozzle; flow structures; linear stability analysis; higher order dynamic mode decomposition

## 1. Introduction

Flow disturbances can provoke severe damage in blood vessels, for instance causing hemolysis [1], in which high-velocity jet streams increase shear forces and disrupt erythrocyte membranes [2]. These disturbances can be caused naturally, with some cardiovascular diseases, like aortic stenosis, or artificially, with the introduction of prosthetic valves or the injection of medicines. Aortic stenosis often causes turbulent flow and significant pressure drops, which have been studied and quantified in previous works [3].

The development of new medical devices, which can prevent the appearance of turbulent flows, would help to overcome the artificially-caused disturbances, reducing the damage that such devices can produce in the human body. With this aim, 'The critical path initiative' program [4] created by the United States (US) Food and Drug Administration (FDA) intends to develop robust and standardized methodologies for medical applications. In this line, a benchmark nozzle (see Figure 1) that resembles an idealized medical device, has been developed with the aim of studying in detail the main characteristics describing the flow inside the geometry. In this geometry, both experiments [5,6] and computational fluid dynamics (CFD) simulations [7–11] have been carried out, remarking the importance of continuing studying this model, using better techniques to model the flow, especially in the turbulent regime [12–15]. Additionally, some authors paid attention to the location of the transition to turbulence and the critical Reynolds number at which such transition occurs, finding that it was influenced by some numerical aspects, such as grid density or numerical noise [16,17]. In experiments, the breakup of the jet is caused by flow instabilities originated by small geometrical disturbances of the domain, however in numerical

simulations, the domain is perfectly symmetric, so the origin of such disturbances was related to some artificial noise [5,15].

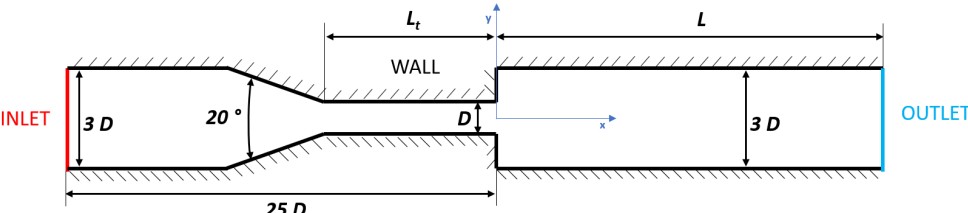

**Figure 1.** Computational domain for the numerical simulations of the planar Food and Drug Administration (FDA) nozzle.

Different inter-laboratory studies [10,11] showed discrepancies among the different turbulent models. In the case with Reynolds number at the throat $Re_t = 500$, the studies do not recommend the use of turbulent models, as experimentally the flow was observed to be laminar. Nevertheless, most of the laminar inlet-cases yielded satisfactory results. Cases with higher Reynolds number ($Re_t = 2000, 3500$) were laminar at the inlet, with a transition to turbulence. In these cases, none of the turbulent models were optimal to recreate the velocities and shear stresses. Although the international standards for implantable circulatory support devices recognize that experimentally validated CFD simulations may be used to characterize the flow fields, the ISO 14708-5 standard indicates that the use of CFD should be limited to the design stage.

Understanding the flow bifurcations, characterized by the main flow patterns driving the flow dynamics, is a good way to start modeling complex flows, providing as result low-order models that are based on the physical insight of the problem studied [18]. These flow patterns can be identified using several techniques. Some of the most popular include: (i) linear stability analysis, which identifies the evolution of a small disturbance upon a base flow (generally steady) that is producing changes in the flow (i.e., flow asymmetries or unsteadiness), and (ii) modal decompositions using data-driven methods, which decompose the flow as an expansion of the highest amplitude modes driving the flow dynamics.

The main objective of this paper is to characterize the flow structures on a planar FDA nozzle using both linear stability analysis, to study the main flow bifurcations occurring at Reynolds number $Re_t \leq 1000$ at the throat (steady flow), and a data-driven method named higher-order dynamic mode decomposition (HODMD) [19], to identify the main flow structures driving the flow dynamics at $Re_t = 3000$ when the flow is unsteady and complex (although still laminar, the flow is two-dimensional so the jet breakdown identified in the literature in three-dimensional configurations is not identified). The linear stability analysis would identify the main mechanism producing changes in the flow (i.e., symmetry breaking, flow unsteadiness), and HODMD will show the mechanisms driving the non-linear flow dynamics, which could be connected to the modes identified by linear stability analysis. However, the origin of turbulence and jet breakdown usually reported in this geometry, which seems to be a quite controversial topic [5,15–17], still cannot be related with the present results, since the flow is two-dimensional and the Reynolds number is still too low. Hence, this remains an open topic for future research. Nevertheless, the present article provides a deeper insight into the flow physics in the FDA nozzle and presents some specific and valuable knowledge regarding the flow instabilities occurring before the turbulent breakdown.

Jotkar and Govindarajan [20] studied in detail the expansion area of the FDA nozzle, using a geometry that was only composed by a short throat and the sudden expansion. These authors carried out a parametric study of the expansion ratio and the divergence angle between the two channels at $Re_t \leq 300$ (steady flow). They performed a linear stability analysis and identified a pitchfork bifurcation, where a leading (steady) mode was breaking the flow symmetry. These results were related to three different topology

regimes: a symmetric flow characterized by two bubbles located in the upper and lower part of the expansion (Regime I), an asymmetric flow (Regime II) and an asymmetric flow with a third bubble identified in the top part of the pipe (Regime III). The flow instability mode was connected with the origin of the second regime (pitchfork bifurcation). These authors also performed a transient growth analysis [21] and the results obtained suggested that there was a connection between Regime III and the optimal energetic mode recovered. In the same line, Wang et al. [22] perform a linear stability analysis to study the same geometry as in [20], but with the flow moving in the opposite direction. In other words, these authors studied a sudden contraction, instead of a sudden expansion. The authors identified the main global mode producing changes in the flow, and used such information to optimize the geometry shape. Nevertheless, the results obtained in the sudden contraction (flow topology and global instability mode), were different from the ones obtained in the sudden expansion.

This work extends the study performed by Jotkar and Govindarajan [20], where the studied geometry is an FDA nozzle instead of a sudden expansion. Linear stability analysis has been performed and the obtained results have been connected with different flow-topology regimes. The three topology regimes and the pitchfork bifurcation identified in the literature [20], has also been identified in the FDA nozzle. Moreover, additional flow regimes have been found, one for the steady flow, and another one when the flow is unsteady and the Reynolds number is intermediate. The present study carried out also identifies a leading unsteady mode that is responsible for the flow unsteadiness. The mode calculated was stable, but the critical Reynolds number defining the flow unsteadiness has been approximated using linear regression. Finally, HODMD has been applied to identify the main patterns in the FDA nozzle in the unsteady regime, providing a deeper knowledge of the flow physics in this geometry, unknown up to now.

This article is organized as follows. Section 2 describes the FDA geometry and the numerical simulations carried out to study the flow inside the FDA nozzle. Section 3 describes the methodology carried out to analyze the flow. Finally, the main results and conclusions are presented in Sections 4 and 5.

## 2. Geometry Description and Numerical Simulations

Numerical simulations have been performed to model the flow inside the two-dimensional Food-and-Drug-Administration (FDA) nozzle. The computational domain of this geometry is presented in Figure 1. As seen, the domain, defined in the $x - y$ (streamwise-normal) plane, is modelled by an inlet tube of diameter $3D$ and length $9.33D$, and a throat of diameter $D = 1$ and length $L_t = 10D$, which are connected by a smooth contraction. Downstream the throat, a sudden expansion leads to a pipe of diameter $3D$ and length $L = 22.5D$. The origin of the reference system is located at the origin of the expansion as presented in the figure. The geometry dimension and model are based on the numerical model as presented in Ref. [12], which reproduces the experimental model in the FDA nozzle.

In the conducted simulations, the flow moves from the left surface (inlet) to the right surface (outlet). As seen in the figure, the top and bottom surfaces of the geometry model the wall of the nozzle. The dynamics of an incompressible Newtonian flow inside an FDA nozzle is governed by the Navier–Stokes equations:

$$\nabla \cdot \mathbf{v} = 0, \tag{1}$$

$$\frac{\partial \mathbf{v}}{\partial t} + (\mathbf{v} \cdot \nabla \mathbf{v}) = -\nabla p + \frac{1}{Re}\Delta \mathbf{v}, \tag{2}$$

where $\mathbf{v}$ and $p$ are the non-dimensional pressure and velocity vector ($\mathbf{v} = (v_x, v_y)$, with $v_x$ and $v_y$ as the streamwise and wall-normal velocity components), and $Re$ is the Reynolds number, defined as $Re = UD/\nu$, where $U$ is the mean velocity at the inlet, $D$ is the throat diameter and $\nu$ the kinematic viscosity of the fluid. The Reynolds number at the throat involves the mean velocity in this area, which is twice the inlet velocity, corresponding to

$Re_t = 2Re$. These two equations have been non-dimensionalized using $D$ as the characteristic length unit and $D/U$ as the time unit.

Numerical simulations have been carried out using the numerical solver Nek5000 (version 19) [23], an open-source code that uses Legendre spectral elements as spatial discretization. This code has been used to accurately simulate a number of complex flow cases in the literature [24,25]. The inlet-velocity boundary condition is modelled using a Poiseuille profile, as:

$$v_x = \frac{2}{3}\left(1 - \frac{y^2}{(3/2)^2}\right), \tag{3}$$

$$v_y = 0, \tag{4}$$

where $y \in [-1.5, 1.5]$, while Neumann boundary conditions are set for the pressure field. For modelling the wall of the geometry, Dirichlet boundary conditions are imposed for both velocity and pressure fields ($v_x = 0$, $v_y = 0$, $p = 0$). Finally, in the outlet surface, for the direction normal to the outflow, the sum of pressure and viscous stress is zero. That means $\nabla \mathbf{v} \cdot \vec{n} = 0$, as $\vec{n}$ is the outward unit normal vector, and Dirichlet for the pressure. The boundary parallel direction is treated as Dirichlet for both velocity and pressure.

The mesh was generated using *Genbox*, the native mesh generator for Nek5000. This mesh contains 8650 spectral elements, each one discretized using eleven Gauss–Lobatto–Legendre points (polynomial degree 10). A larger concentration of elements is located in the area of the sudden expansion, to guarantee the good resolution of the results.

An explicit second-order extrapolation scheme for the time integration has been used as the temporal discretization. A grid independence study has been carried out in order to observe the influence of the polynomial degree on the simulations. Polynomial degrees 8, 10 and 12 have been used to solve the problem, comparing the streamwise and spanwise velocity components at 10 representative points along the domain. This study has been carried out at Reynolds numbers 500, where the flow converges to a steady state, and 1500, where the flow is unsteady. The variations in the velocity components in the results obtained at Reynolds number 500 were found between the fifth and sixth significant decimal digit, so the three cases compared show similar results. At Reynolds number 1500, larger differences were identified between the solution with polynomial degree 8, and with polynomial degrees 10 and 12. Nevertheless, the largest differences in the velocity vector identified between the results obtained with polynomial degrees 10 and 12 were found between the second and third significant decimal digit. For simplicity (reduced computational cost), the solution with polynomial degree 10 is selected to proceed with the data analysis carried out in this article, assuming a relative error of ∼2–3% in the results presented. This error is calculated comparing the L2 norm of the velocity vector in the results obtained with polynomial degrees 10 and 12, considering the latter case as the reference. Moreover, following the results presented in previous works [20,26–28], the base flow has been validated comparing the main topology patterns describing the flow (the separation bubble, described in detail below, in Section 4.1). As in the literature, the present article compares the evolution of the size of the two bubbles as a function of the Reynolds number. More specifically, it shows the position of the reattachment point of the flow. Figure 2 shows that the numerical results from this article agrees with the literature.

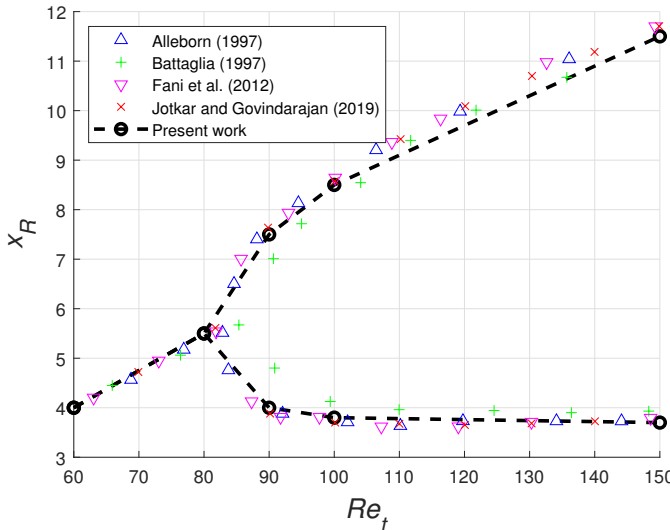

**Figure 2.** Reattachtment point $x_R$ of the separation bubble identified as main topology patterns in the FDA geometry as a function of the Reynolds number at the throat ($Re_t = 2Re$). Comparison of the present numerical simulations with the literature [20,26–28].

## 3. Methodology

This section introduces the two methodologies employed to identify the main patterns and bifurcations in the FDA nozzle: linear stability analysis and higher-order dynamic mode decomposition. The first technique is used to identify the main bifurcations occurring in the flow at low Reynolds number, when it is steady. The second method is a data-driven technique that has been used to identify the main flow structures modeling the unsteady flow in the FDA nozzle at intermediate Reynolds number.

### 3.1. Linear Stability Analysis

Linear stability theory (LST) studies the evolution of a small perturbation imposed upon a base flow. Following the Reynolds decomposition, it is possible to decompose the instantaneous flow field, defined by the state vector representing the velocity vector and pressure fields as $\mathbf{q}(\mathbf{x}, t) = (\mathbf{v}(\mathbf{x}, t), p(\mathbf{x}, t))$, as follows:

$$\mathbf{q}(\mathbf{x}, t) = \mathbf{Q}_b(\mathbf{x}, t) + \hat{\mathbf{q}}(\mathbf{x}, t), \tag{5}$$

where $\mathbf{Q}_b(\mathbf{x}, t)$ is the base flow and $\hat{\mathbf{q}}(\mathbf{x}, t)$ represents the flow perturbations. The classical theory defines the base flow as a steady state [29,30], although in more recent contexts, the base flow is also considered as the mean flow from unsteady solutions [31]. The present article focuses on studying the first flow bifurcations, hence the base flow is a steady solution.

Introducing (5) into the Navier–Stokes Equations (1) and (2), and linearizing them over the base flow, it is possible to obtain the linear form of the Navier–Stokes equations, defined as:

$$\nabla \cdot \hat{\mathbf{v}} = 0, \tag{6}$$

$$\frac{\partial \hat{\mathbf{v}}}{\partial t} + \hat{\mathbf{v}} \cdot \nabla \mathbf{V}_b + \mathbf{V}_b \cdot \nabla \hat{\mathbf{v}} = -\nabla p + \frac{1}{Re} \Delta \hat{\mathbf{v}}. \tag{7}$$

Writing these equations as an initial value problem and taking into account the separability between temporal and spatial coordinates, where the following Fourier decomposition is introduced in time as $\hat{\mathbf{q}} = \tilde{\mathbf{q}} e^{-i\lambda t}$, we obtain the following generalized matrix eigenvalue problem

$$\mathbf{B}\tilde{\mathbf{q}} = \lambda \mathbf{A}\tilde{\mathbf{q}}, \tag{8}$$

with **A** and **B** being the matrices that collect information regarding the boundary conditions of the problem. The eigenvectors $\tilde{\mathbf{q}}$ define the shape of the most unstable modes leading the flow motion, which are identified by means of the eigenvalues $\lambda$, defined as $\lambda = \sigma + i\omega$, where $\omega$ is the oscillation frequency and $\sigma$ is the growth rate. For values of $\sigma > 0$, the solution of the system is unstable, producing changes in the flow (i.e., symmetry breaking, flow unsteadiness, flow three-dimensionality), while for values of $\sigma < 0$, this solution is stable and the flow remains as initially. Two types of solutions are possible. In the first case, the flow is unstable, but the mode is steady ($\omega = 0$), while in the second case the mode oscillates with a frequency $\omega \neq 0$. These are known as pitchfork and Hopf bifurcations, respectively. The present paper identifies these two types of bifurcations, originated by the presence of some artificial noise (small perturbation) in the base flow, growing in time and producing changes in it. In the first case, when the leading mode becomes unstable as a function of the rise in the Reynolds number (passes from $\sigma < 0$ to $\sigma > 0$), the base flow modifies its shape, leading into a new base flow (also steady), while in the second case, the base flow becomes unsteady. Studying further flow bifurcations would require more complex analyses, such as Floquet stability analysis [32], which is beyond the scope of this paper. Nevertheless, more complex flows (formed by a larger number of frequencies) can be analyzed using more sophisticated techniques as introduced in the following section.

The linear stability analysis has been carried out using the module for linear stability analysis of the solver Nek5000 [23], which uses the Arnoldi method implemented in the software package ARPACK. More details about the numerical implementation of the code can be found in Ref. [33]. The boundary conditions for the computation of the base flow are the same as for the non-linear numerical simulations: the velocity inlet is modeled with the Poiseuille profile as in Equations (3) and (4), no-slip boundary conditions are used for the walls of the geometry and outflow conditions are used for the outlet surface, namely, for the direction normal to the outflow, the sum of pressure and viscous stress is zero. That means $\nabla \mathbf{v} \cdot \vec{n} = 0$, as $\vec{n}$ is the outward unit normal vector, and Dirichlet for the pressure. The boundary parallel direction is treated as Dirichlet for both velocity and pressure. The mesh used to solve the analysis is the same as in the numerical simulations for the non-linear Navier–Stokes equations, previously described in Section 2. Hence the mesh is composed of 8650 elements and the polynomial order for the Legendre spectral elements is 10. The results obtained in this article are in good agreement with the results presented in Ref. [20], where they perform the linear stability analysis to study a sudden expansion geometry (same as the FDA but without the convergent part of the channel). More details will be presented in Section 4.2.

### 3.2. Higher-Order Dynamic Mode Decomposition

Higher-order dynamic mode decomposition (HODMD) [19] is a data-driven method used in flow dynamics to identify flow patterns. HODMD is an extension of dynamic mode decomposition (DMD) [34] introduced for the analysis of complex flows, such as noisy experiments [35,36], transitional flows [37], turbulent flows [38], et cetera, to either study the flow physics [39] or to create reduced order models [40,41].

Similarly to DMD, HODMD decomposes the original data $\mathbf{v}_k$ (snapshot) as an expansion of DMD modes in the following way:

$$\mathbf{v}_k \simeq \mathbf{v}_k^{DMD} \equiv \sum_{m=1}^{M} a_m \mathbf{u}_m e^{(\delta_m + i\omega_m)(k-1)\Delta t}, \quad k = 1, \ldots, K, \tag{9}$$

where $\mathbf{u}_m$ are the DMD modes, and $\omega_m$, $\delta_m$, $a_m$ are their corresponding frequencies, growth rates and amplitudes, respectively. The number of terms, $M$, can be referred to as the spectral complexity and $K$ as the temporal dimension (generally $K$ represents the number of snapshots in the data analyzed). The number of DMD modes retained in the expansion

($M$) depends on a tolerance $\varepsilon_1$, which is a parameter tunable as a function of the desired accuracy to reconstruct the original data using the previous DMD expansion.

The algorithm is described in detail in Ref. [19] (see Ref. [42] to download the code and to find more details about the way of applying the method). This algorithm can be summarized in two main steps. Initially, the data are collected into a snapshot matrix. The first step applies a singular value decomposition (SVD) to the initial data with the aim of reducing the spatial redundancies, removing the noise (in experiments), or retaining the large size flow structures (in turbulent flows). A second tolerance (tunable) $\varepsilon$ defines the number of SVD modes retained, representing the main flow dynamics. A DMD-like algorithm is then applied to this (clean) data in the second step. This algorithm uses $d$ time-delayed snapshots from the snapshot matrix, resulting in a similar process to the sliding window from the power spectral density (PSD) analysis. This second step is called the DMD-$d$ algorithm, where $d$ is a tunable parameter. At this step, the tolerance $\varepsilon_1$ (previously mentioned) defines the number of $M$ DMD modes retained in the expansion (9).

HODMD will be applied to identify the main patterns in the non-linear solution of the numerical simulations carried out in the FDA geometry. The high complexity of the data analyzed (a large number of involved frequencies) makes HODMD (a more robust approach of DMD [39] introduced for the analysis of complex flows [36,38]) a suitable tool to identify the main frequencies driving the flow dynamics.

## 4. Results

### 4.1. Flow Topology

The evolution of the main topology patterns in the FDA nozzle has been studied in detail in the solution of the numerical simulations obtained at different Reynolds numbers ranging in the interval $Re \in [30, 500]$. At this conditions the flow is steady, hence the solutions presented below will be used as the base flow for the linear stability analysis that is performed in the following section. Figure 3 shows the evolution of the streamwise velocity and the streamlines in the results obtained (all represented by a laminar flow, let us remember that the flow presented in this article is two-dimensional, hence we do not expect the jet breakdown, connected with three-dimensional effects occurring in the azimuthal component). As in the work presented by Jotkar and Govindarajan [20], it is possible to identify three different regimes, although, as mentioned before, the geometry studied in the present article is slightly different from the one presented in the literature, which is a geometry only composed by the throat and the expansion regions of the FDA nozzle (the smooth contraction is missing).

In Regime I ($Re = 30$ and $40$) the flow is symmetric. Two identical bubbles are identified in the top and bottom corners of the expansion of the geometry. At $Re = 45$ ($Re_t = 90$), the flow is not symmetric anymore and the size of the bubbles change: the bubble of the top shrinks while the bubble of the bottom increases its size. This symmetry breaking defines the origin of Regime II. The critical Reynolds numbers distinguishing regimes I and II is $Re_c \simeq 42.5$ (in between $Re = 40$ and $Re = 45$), in good agreement with the literature [20], where this critical Reynolds number was estimated as $Re_t \simeq 80$ ($Re \simeq 40$) by linear stability analysis.

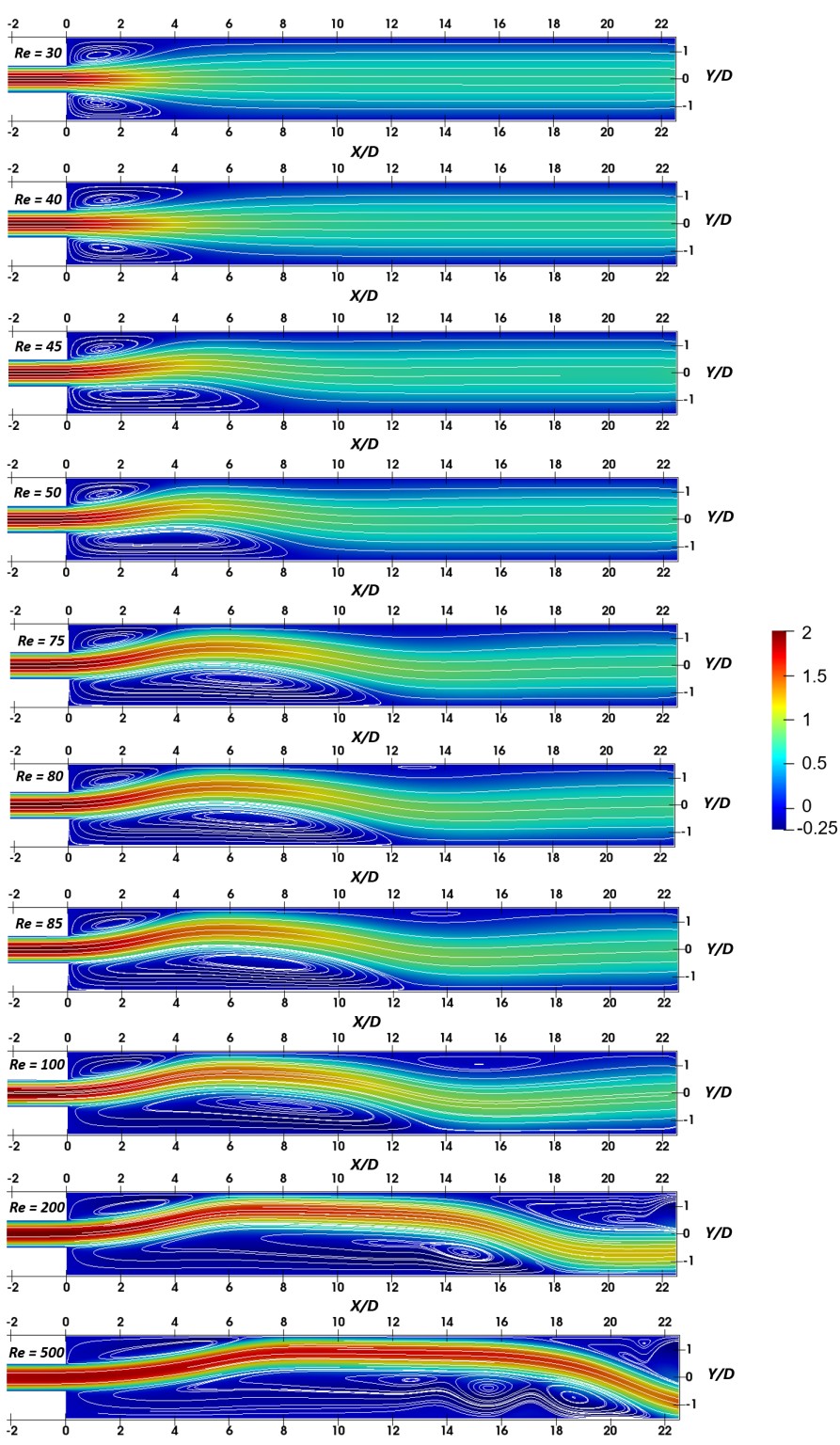

**Figure 3.** Streamwise velocity (color map presented on the right figure) and streamlines showing the evolution of the flow field at different Reynolds numbers (marked as *Re* in each panel). Zoomed-in view for $x \in [-2, 22.5]$.

The presence of a third small bubble in the top part of the domain, in the middle part of the pipe, at $Re \simeq 80$ ($Re_t \simeq 160$), defines the origin of Regime III. For simplicity, this small bubble is called as top-middle bubble. The critical Reynolds number that differentiates Regimes II and III is $Re_c \simeq 75$ ($Re_t \simeq 150$), again in good agreement with the results presented in Ref. [20]. The size of the bottom and top-middle bubbles increases with the

Reynolds number, while the small bubble of the top corner of the expansion maintains its size since the beginning of Regime II. Additionally, the top-middle bubble moves forward for larger Reynolds numbers. At $Re = 200$ ($Re_t \simeq 400$) it is possible to identify a new flow regime, i.e., Regime IV. The critical Reynolds number defining the origin of this regime is located at some point defined in the interval $100 < Re_c < 200$. In Regime IV, the small bubble of the top corner also starts increasing its size. Simultaneously, the bottom bubble slightly changes its topology, stretching out and splitting into two parts, giving rise to the presence of a new bubble, as it is confirmed at $Re = 500$, where the bottom bubble is divided into three small bubbles. Moreover, a new bubble is identified adjacent to one of these three bubbles, reaching the bottom of the pipe. The higher complexity of the flow in this area suggests that new flow bifurcations could promote the presence of vortex shedding in this part of the geometry at higher Reynolds numbers, which are related to the unsteadiness of the flow, as it will be presented in Section 4.3. Regarding the top-middle bubble, at $Re \geq 200$ this bubble moves towards the output of the domain when the Reynolds number increases, probably vanishing for larger values of $Re$. The size of the three bubbles have been compared with the previous work [20], measuring the streamwise extent of the recirculation zones as a function of the Reynolds number (see Figure 2).

Finally, Figure 4 shows the streamwise velocity field and the streamlines in the flow at $Re = 1500$ ($Re_t = 3000$).

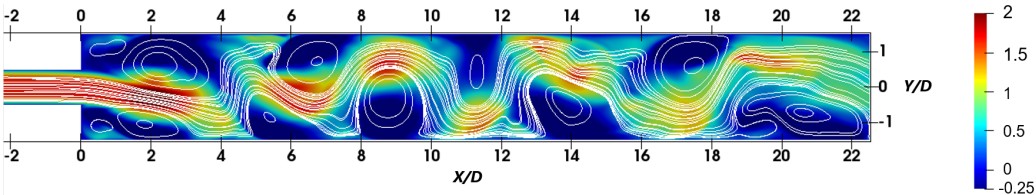

**Figure 4.** Same as Figure 3 for $Re = 1500$ ($Re_t = 3000$). Representative snapshot of the unsteady flow.

In these conditions, the flow is unsteady, hence the figure shows a representative snapshot. As can be observed, in this regime the three bubbles identified in the previous regimes are missing. The steady flow presented in the previous regimes breaks into a vortex street, which is distributed through all the pipe.

*4.2. Linear Stability Analysis*

The linear stability analysis has been performed to study the evolution of the growing small disturbances on the flow. The linear global modes have been calculated at different Reynolds numbers, identifying the critical Reynolds number at which the flow becomes unstable, and consequently changes its initial state.

In the conducted calculations, four different modes were identified, i.e., two steady modes, and two unsteady modes, which are stable in the calculations performed, although the critical Reynolds numbers at which these modes will become unstable have been approximated by means of linear regression. The steady modes identified are related with variations in the base flow, while one of the two identified unsteady modes will be responsible for the transition from steady to unsteady flow. Figure 5 top shows the evolution of the growth rate of the four leading modes identified as a function of the Reynolds number, while the bottom figure shows the evolution of the frequency of the two unsteady modes. For simplicity, the two steady modes are called as mode A and mode B, while the two unsteady modes are the mode 1 and mode 2.

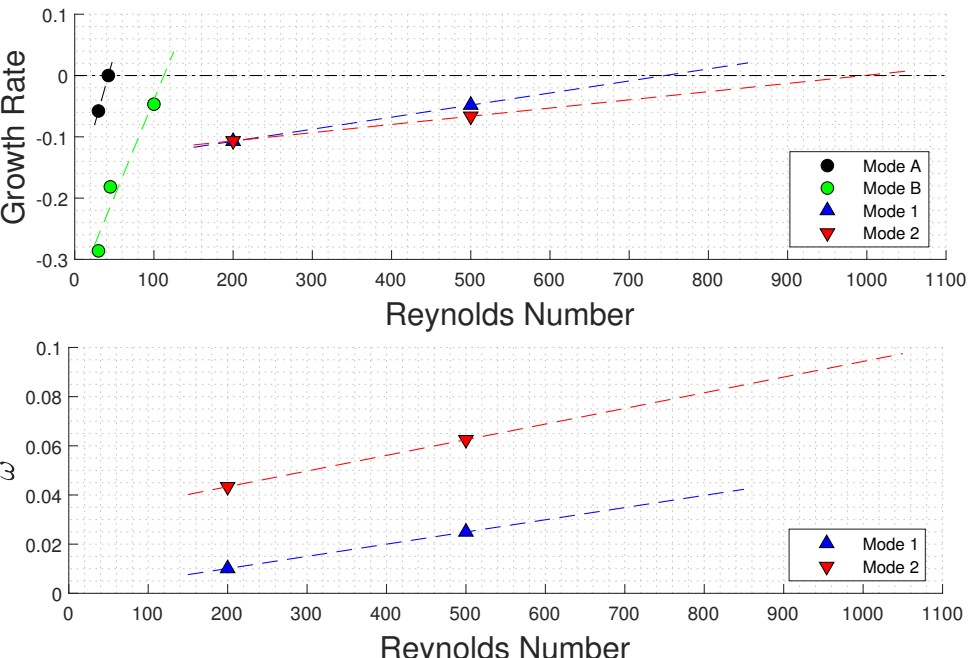

**Figure 5.** Growth rate (**top**) and frequency (**bottom**) vs. Reynold number for the eigenvalues associated with the four leading modes identified in the linear stability analysis.

Starting from the top figure, which follows the evolution of the growth rates, it is possible to see that the first flow bifurcation is driven by mode A, which is steady. This mode becomes unstable at $Re \simeq 40$ ($Re_t \simeq 80$), which is connected to the symmetry breaking, detailed in the topological changes presented Figure 3. The mechanism producing this flow instability is a pitchfork bifurcation. This result is in good agreement with the literature [20], as well as the evolution of mode B, which is always stable in the calculations performed, since it is recovered using a time stepper code. Following a linear regression tendency to extrapolate the results obtained, the critical Reynolds number of mode B, mode 1 and mode 2 is ∼130, ∼740 and ∼1000. Although at $Re = 200$ the growth rate of mode 1 and mode 2 is similar, following the tendency of these modes at larger Reynolds numbers it is found that mode 1 is more unstable than mode 2, hence it is expected that mode 1 would be the main mechanism triggering the flow unsteadiness. Figure 5, bottom, shows the evolution of the frequency as a function of the Reynolds number in mode 1 and mode 2. As observed, the frequency of mode 1 is smaller than that frequency of mode 2, suggesting that the main changes produced in the flow would be driven by a low-frequency mode. In the figure, the frequencies of modes 1 and 2 are extrapolated using a linear regression to estimate the frequency of these modes at their corresponding critical Reynolds number. Following this tendency, the frequency of mode 1 at $Re \simeq 740$ is $\omega \simeq 0.038$ and for mode 2 at $Re \simeq 1000$ is $\omega \simeq 0.095$.

Figure 6 shows the streamwise velocity components and the streamlines of mode A calculated at different Reynolds numbers, before and after the critical Reynolds number of this mode. As mentioned above, this mode is responsible for the symmetry breaking produced in the flow. The shape of this mode is in good agreement with the results presented by Jotkar and Govindarajan [20].

The highest activity of mode A is located near the expansion region. The mode is symmetric below the critical Reynolds number, although the symmetry breaks for values above such Reynolds number. The changes in the shape of the mode and its location justifies once more that this mode is the main mechanism triggering the symmetry breaking shown in Figure 3, where the bottom bubble increases its size. This is also in good connection with the shape of this mode at $Re = 45$, where the lower part of the mode presents a small area

where the intensity of its activity is attenuated in the region where the large size bottom bubble would be located.

The mode B is presented in Figure 7. The highest intensity of mode B is located in the mid part of the channel, in $8.5 \leq x/D \leq 18$.

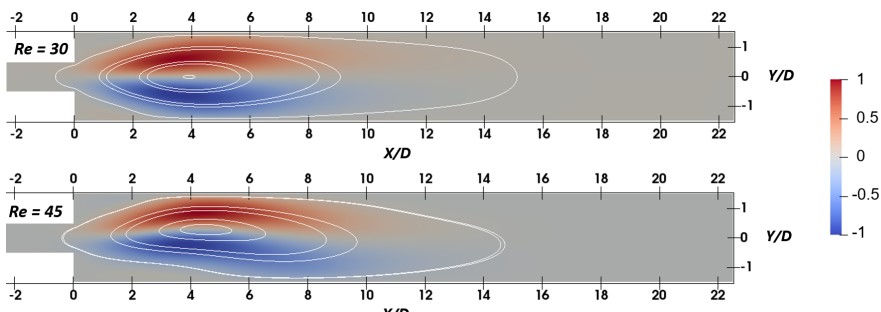

**Figure 6.** Streamwise velocity component (color map presented on the right figure) and streamlines in mode A for $Re = 30$ (**top**) and $Re = 45$ (**bottom**). Mode A becomes unstable at $Re = 40$. Zoomed-in view for $x \in [-2, 22.5]$.

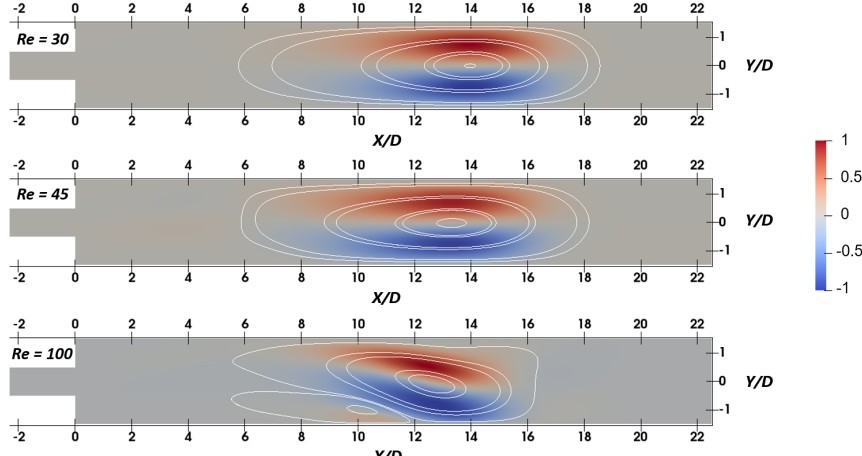

**Figure 7.** Streamwise velocity component (color map presented on the right figure) and streamlines in mode B for $Re = 30$ (**top**), $Re = 45$ (**center**) and $Re = 100$ (**bottom**). The mode is always stable at the present flow conditions. Zoomed-in view for $x \in [-2, 22.5]$.

This mode is symmetric with respect to the mid part of the channel at $Re = 30$ and 45 (below and above the critical Reynolds number of the first flow bifurcation). At $Re = 100$ the mode inclines itself, although the main activity is maintained in the same area as before. At this flow conditions, the streamlines of the mode show a small bubble adjacent to the mode, reaching the bottom wall of the pipe. Although the shape of the mode changes with the Reynolds number, the regime identified in Figure 3 corresponds to Regime III in the three cases presented. This suggests that the changes in the shape of the mode are connected with the larger size and the position of the top-middle bubble, which travels downstream as Re increases, leading to the new Regime IV shown in Figure 3 at $Re = 200$.

Finally, Figure 8 shows mode 1, the most unstable of the two unsteady modes identified.

The main activity of the mode is in the mid part of the channel, in both cases just upstream the main core of the bottom bubble from Figure 3. The region of highest activity of the mode at $Re = 200$ is smaller than at $Re = 500$, suggesting that the influence of the mode over the entire flow field increases when it becomes more unstable. This suggests the connection of this mode with the origin of the vortex street identified in the entire length of the pipe at $Re = 1500$, as previously presented in Figure 4.

A parametric study has been carried out, changing the value of the Krylov subspace, with values of 1000, 2000 and 4000. The variations in the eigenvalues among the three cases were found between the sixth and seventh decimal digit, so did the eigenmodes.

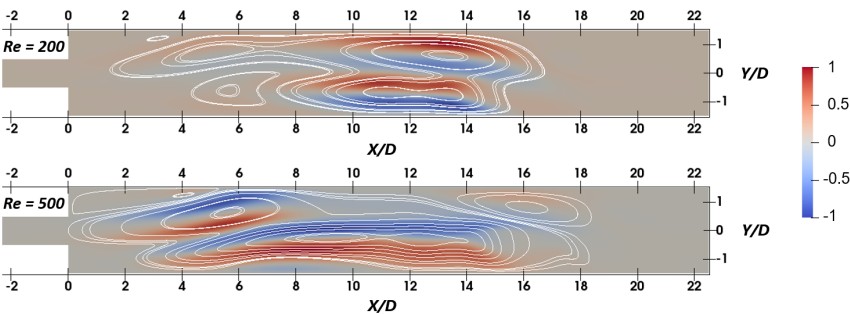

**Figure 8.** Streamwise velocity component (color map presented on the right figure) and streamlines of mode 1 for $Re = 200$ (**top**) and $Re = 500$ (**bottom**). The mode is always stable at the present flow conditions. Zoomed-in view for $x \in [-2, 22.5]$.

*4.3. Flow Structures in the Unsteady Flow*

This last section identifies the main structures in the flow presented in Figure 4. This flow has been obtained solving the non-linear Navier stokes equations at Reynolds number 1500 ($Re_t = 3000$), resulting in an unsteady flow. As described in Section 4.1, the topology of this flow is quite different to the one presented at low Reynolds number, when the flow was steady. Hence the global modes identified in this case may differ from the ones identified in the previous section by linear stability analysis.

HODMD has been applied to analyze a set of 600 snapshots equidistant in time with $\Delta t = 0.3$. To ensure that the flow was saturated (the numerical simulations were converged), avoiding in this way the presence of transitional (growing or decaying) modes, these snapshots were collected after running the code for 1000 time units. Hence, the data analyzed are contained in the time interval $t \in [1001, 1181]$.

HODMD was calibrated first to ensure the high accuracy of the results presented. Hence, various analyses were performed using the tolerances $\varepsilon = \varepsilon_1 = 10^{-2}$ and $5 \times 10^{-2}$, which provide more and less accurate results, respectively, and the parameters $d = 100$, 120 and 150. Figure 9 shows the frequencies as a function of the amplitudes of the DMD modes identified with the different tolerances. The top figure shows all the results obtained. As can be observed, the spectrum is noisy, and it is difficult to identify the leading modes.

The top part of the figure shows a zoomed-in view for the low-frequency modes. The less accurate tolerance, $\varepsilon = \varepsilon_1 = 5 \times 10^{-2}$, retains a smaller number of modes with frequency smaller than $\omega = 2$ in all the cases. In this figure it is possible to identify only a few modes that are robust. The robust modes are the ones presenting similar frequencies and amplitudes in all the tests carried out, forming clusters of modes. Based on previous work [38] these modes can be considered as the physical modes driving the flow. The frequency of the robust modes is $\omega = 0$, for the one with the highest amplitude, representing the mean flow, and $\omega = 0.35$, $\omega = 0.5$ and $\omega = 0.93$ for the unsteady remaining robust modes. Let us note that in the spectrum it is possible to identify a larger number of modes considering only the results with the more accurate tolerance $\varepsilon = \varepsilon_1 = 10^{-2}$, which would complement the results presented. Nevertheless, the present article only intends to identify the highest amplitude modes to give a general description of the main patterns driving the flow. Studying in detail the spectrum and identifying more relevant modes describing the flow remains as an open topic for future research, with the aim of using this information to create novel accurate reduced-order models based on the physical insight of the flow.

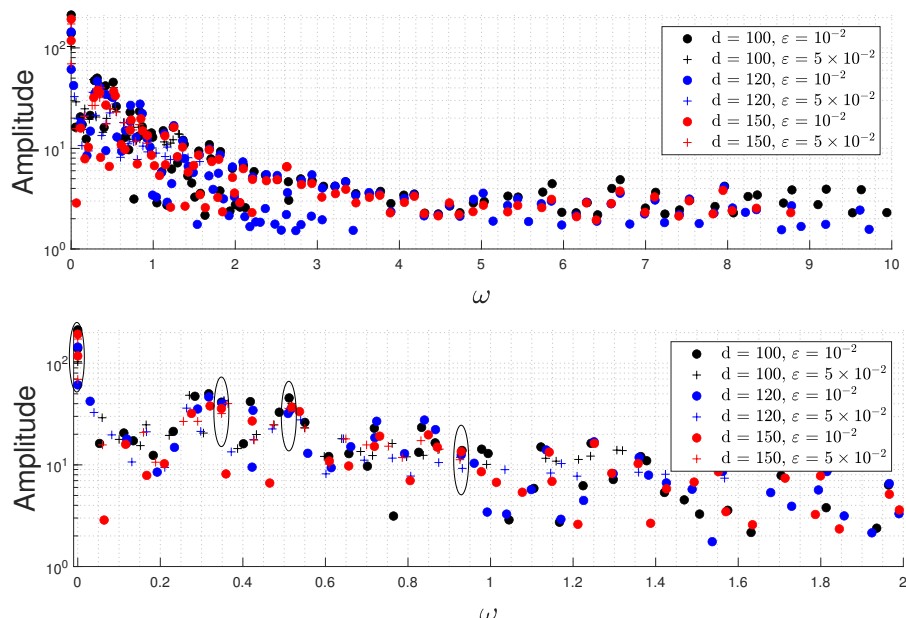

**Figure 9.** Frequencies vs. amplitudes representing the dynamic mode decomposition (DMD) modes in the plannar FDA nozzle. **Top**: complete spectrum. **Bottom**: blow up of the region with frequencies between 0 and 2.

Figure 10 shows the real part of the streamwise component of the main DMD modes.

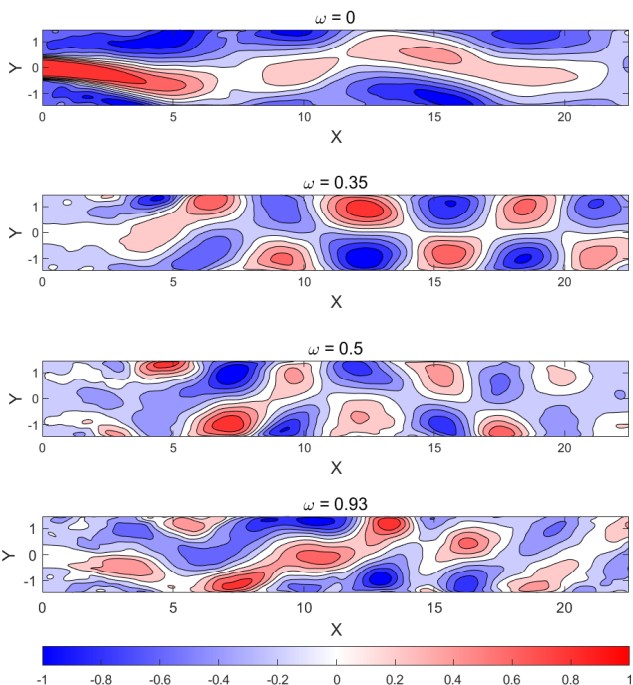

**Figure 10.** Real part of the streamwise velocity component (color map presented on the bottom figure) of the DMD modes. From top to bottom: $\omega = 0, 0.35, 0.5$ and $0.93$. The modes are normalized with their maximum value. Zoomed-in view for $x \in [0, 22.5]$

The mode representing the mean flow is asymmetric, in good agreement with the first flow bifurcation studied by means of the linear stability analysis in the previous section. The remaining modes describe a vortex street whose main activity starts when the main activity of the steady mode finishes. Although, depending on the frequency of the

mode, the main activity and the shape of the vortex street is slightly different. These three unsteady modes describe the main dynamics driving the flow in the vortex street identified in the expansion of the FDA nozzle presented in Figure 4. Controlling the presence of these modes (and also the presence of the linear stability modes, which are identified first), could aid the construction of new medical devices that minimize their damage produced in the human body. However, this remains as an open topic for future research.

## 5. Conclusions

This article studies in detail the main flow patterns and bifurcations described in the FDA nozzle, a benchmark nozzle that resembles an idealized medical device introduced in 'The critical path initiative' program [4] created by the United States.

Numerical simulations have been performed at several Reynolds numbers using the solver Nek5000 [23], obtaining steady solutions with different flow topologies in the expansion of the geometry. The results obtained have been validated with the literature [20]. Depending on the flow topology, four different regimes have been identified as a function of the Reynolds number. Starting from low to high Reynolds number it is possible to define Regimes I–IV. In Regime I, the flow is symmetric. This regime is characterized by the presence of two small bubbles located in the upper and bottom corners of the expansion. Regime II is introduced by a symmetry breaking, where the upper bubble shrinks while the bottom bubble enlarges. In Regime III, a third bubble is identified in the upper part of the expansion, in the middle part of the pipe, reaching the top wall. Finally, at higher Reynolds number it is possible to identify Regime IV, where the bubble of the upper–middle part moves downstream, partially leaving the domain, and the bottom bubble splits into several small bubbles. These topology patterns have been compared with other results obtained at larger Reynolds numbers, when the flow is unsteady. In these conditions a vortex street is identified distributed along the entire length of the pipe, starting on the expansion area.

Linear stability analysis has been performed to identify the leading flow bifurcations and its connection with the changes in the topology of the flow. From this analysis, three leading modes have been identified: two steady modes and one unsteady mode. The first leading mode is steady. This mode is considered as the main mechanism triggering the symmetry breaking previously described, considered as the origin of Regime II. The mode represents a pitchfork bifurcation. The second leading mode is also steady. This mode is always stable in the calculations performed, but the critical Reynolds numbers that define the stability limit of this mode have been estimated extrapolating the results obtained using linear regression. This second mode has been related to the presence of the top bubble identified in the topological Regime III. Finally, the third mode is a low-frequency mode, which is the main mechanism triggering the flow unsteadiness. This mode is always stable in the conducted calculations, although as in the previous case, the critical Reynolds number defining the origin of the flow unsteadiness is estimated by linear regression. The highest intensity of this mode is located upstream the small bottom bubbles defined in Regime IV.

Finally, the unsteady solution of the flow calculated at high Reynolds numbers has been studied in detail. The main flow patterns have been identified using HODMD, which provides the leading DMD modes describing the flow. The flow analyzed is composed of a vortex street distributed through all the pipe defining the expansion of the geometry. This topology is totally different from the steady laminar flow studied with the linear stability analysis, therefore, the DMD modes identified by the method are different from the ones of the linear analysis. HODMD identifies four leading modes, which model the main dynamics of the flow. The most relevant mode is steady, representing an asymmetric solution (the mean flow), in good agreement with the first linear stability mode found at low Reynolds number, which was triggering this symmetry breaking. The three remaining modes are unsteady, and represent the vortex street distributed along the pipe. These modes present a similar shape, although their region of highest intensity slightly changes among them. HODMD also identifies other relevant modes, which are not studied in

detail in the present paper, because their (spatial) amplitude is small. This paper intends to identify the driving dynamics of the flow field, rather than creating a reduced-order model of the flow. Nevertheless, studying in detail this solution remains an open topic for future research.

**Author Contributions:** A.C. performed the numerical simulations and the data analysis with the support of D.X. A.C. wrote the article with the support of S.L.C., R.V. and P.S., who also developed the idea. All authors have read and agreed to the published version of the manuscript.

**Funding:** This study was also funded by the Swedish Foundation for Strategic Research, project "In-Situ Big Data Analysis for Flow and Climate Simulations" (ref. number BD15-0082) and by the Knut and Alice Wallenberg Foundation.

**Acknowledgments:** A.C. and S.L.C. acknowledge the support of the Spanish Ministry of Education and Vocational Training by the award of a collaboration scholarship.

**Conflicts of Interest:** The authors declare no conflict of interest.

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
