# Peer review of "Flow Structures on a Planar Food and Drug Administration (FDA) Nozzle at Low and Intermediate Reynolds Number"

_fluids, doi:10.3390/fluids6010004_

Round 1

Reviewer 1 Report

The presented study is focused on the stability analysis of the flow structures on a planar FDA nozzle based on the CFD results.

The introduction sufficiently describes the state of the art in the presented field. The goals of the paper are here well defined.

The paper has a logical structure with the well-described numerical methods, geometry and methodology of the data processing. Conclusions are relevant.

I would like to address the authors with the following note and questions:

There are several recent papers dealing with the numerical simulation of the FDA nozzle which report that the prediction of the jet breakup distance by CFD  is strongly influenced by the factors such as grid density, the order of discretization and other aspects of the numerical simulation setup [1, 2]. The problem is caused by the fact that in experiments is the breakup of the jet caused by the instabilities originating in the small geometrical disturbances of the domain, while in CFD the disturbances have the numerical origin (as the computational domain is perfectly symmetric). The better coincidence with the experimental results was achieved by introducing small artificial disturbances (minor random angular velocities for example).

[1] https://onlinelibrary.wiley.com/doi/full/10.1002/cnm.3150

 Questions:

1- Have you involved such "artificial noise" into your simulations?

2 - Do you have a comparison of your results with the experimental data to support your conclusions?

I recommend the paper to publish after answering the given questions.

Round 2

Reviewer 2 Report

The authors have elaborated and have taken care of all the concerns that I raised. I congratulate the authors for an excellent work and recommend this paper for publication in Fluids.